# Improved AIS data simplification algorithm for extracting typical routes considering motion continuity

Jin He[1,2], Jinjia Ruan[1*], Yao Tong[1]

1 Marine Navigation Support Technology Research Center, China Waterborne Transport Research Institute, Beijing, China, 2 School of Information Science and Technology, Dalian Maritime University, Dalian, China

* ruanjinjia@wti.ac.cn

## Abstract

Automatic Identification System (AIS) data provides crucial information about vessel trajectories. However, raw AIS data is often highly redundant, containing overlapping and repetitive routes, which complicates its direct use in maritime applications such as navigation planning and route prediction. In this paper, we propose an improved simplification algorithm designed to extract typical routes while preserving vessel movement continuity. Our approach simplifies complex AIS data by applying an enhanced distance threshold pruning technique and analyzing the continuity of vessel operations to address route segment discontinuities and coordinate deviations. We conducted experiments to evaluate the impact of the simplification algorithm on deep learning applications, specifically in trajectory prediction and anomaly detection. The results demonstrate that the simplified data significantly improves both training efficiency and prediction accuracy in trajectory forecasting models using deep learning, while also enhancing anomaly detection capabilities. Compared to models trained on the original AIS data, those trained on the simplified data achieved faster convergence and higher precision, with fewer false positives in anomaly detection tasks. The findings highlight the practical advantages of the proposed simplification method, making it a valuable tool for real-time maritime monitoring and improving overall operational efficiency. Our code and data at https://doi.org/10.5281/zenodo.17568672.

## Introduction

The Automatic Identification System (AIS) is a navigation aid system that transmits extensive data reflecting the movement status of vessels. Installed on ships, the AIS system periodically sends information such as position, speed, and heading, enabling automatic identification and information exchange between vessels [1].

AIS data has widespread applications in various fields, including port and waterway management [2], collision risk assessment [3], route planning [4], and marine

**Data availability statement:** The code and anonymized dataset supporting this study have been deposited in Zenodo with the DOI: https://doi.org/10.5281/zenodo.17568672.

**Funding:** This work was supported in part by the Basic Scientific Research Operating Expenses Project (grant code: 132505) from China Waterborne Transport Research Institute. The funder is He J (initials of He Jin). The sponsors/funders had no role in the study design, data collection and analysis, decision to publish, or preparation of the manuscript.

environmental protection [5], and navigational safety enhancement under complex weather conditions [6]. For instance, port authorities utilize AIS data to monitor vessel movements in and out of ports, thereby enhancing operational efficiency and safety. As technology advances and applications deepen, AIS data can be leveraged to analyze the behavioral patterns of ships, most notably reflected in their trajectories [7].

However, in practical applications, various factors such as the marine environment, weather conditions, navigational areas, and vessel missions can lead to significant deviations in the trajectories of vessels within the same voyage [8]. Additionally, AIS data is collected at specific time intervals, which may result in missing attribute data [9]. Furthermore, due to the high density and extensive length of the collected AIS data, the direct use of this data often results in overlapping, backtracking, and redundant navigation paths.

Currently, data compression algorithms are the primary method for simplifying AIS data. However, these algorithms generally perform well for individual voyages but may retain unnecessary features when dealing with repetitive and redundant trajectories. Therefore, there is a need to further extract typical routes, which can remove redundant data and preserve the overall navigation pattern, providing a clearer representation of vessel routes.

The extraction of typical routes thus holds notable practical significance, as it provides a concise reference for trajectory prediction [10], facilitates the detection of deviations thereby enhancing safety, supports efficient route planning, and enables the analysis of navigational strategies under diverse environmental scenarios [11]. However, applying existing AIS data simplification algorithms, such as the Sliding Window (SW) and its variants (e.g., IOPW, DTGO), directly to this task of deriving typical routes from long-term cumulative data presents certain challenges. Their design typically assumes the presence of clearly defined start and end points for a single voyage, which may not be explicitly available in continuous, multi-voyage data streams. Furthermore, their performance often benefits from the availability of complete kinematic parameters, meaning their effectiveness can be diminished when such data is incomplete, as is common in practical AIS records. Most importantly, their core objective is to preserve the geometric shape of a single trajectory during compression, which differs from the goal of distilling a set of multiple, representative routes from a collection of overlapping voyages. Consequently, when applied to cumulative data, these methods may retain the inherent redundancy across numerous voyages rather than filtering it out to generate the concise set of typical routes.

In summary, we propose an algorithm for extracting typical routes from accumulated AIS data for a single vessel. The algorithm first leverages the temporal continuity of AIS data and combines an improved distance threshold pruning algorithm to compress trajectory data, resulting in multiple candidate routes. Subsequently, it analyzes the continuity of vessel operation and movement patterns to optimize errors in route discontinuities and intersection points, merging route segments to ultimately generate several intuitive and refined typical route segments.

The contents of this paper are organized as follows. We review the related work in Section 2 and introduce our algorithm in Section 3. Then, the results of experiments to validate our algorithm are exhibited in Section 4. Finally, we draw conclusions in Section 5.

## Related work

A standard route includes specific starting points, destination points, and key waypoints, acting as a guide for planning ocean voyages for most ships. Research shows that AIS data may contain multiple typical routes, which can be seen as real routes influenced by marine environment, weather, and other factors, relative to these standard routes [12]. These typical routes hold significant research value. Currently, extracting typical routes from AIS data primarily involves data simplification algorithms [13].

One simplification algorithm is the Dougla-Peucker (DP) algorithm [14]. This algorithm's core involves connecting the start and end points of the initial trajectory, calculating the perpendicular distance of all intermediate points to this line, and comparing these distances against a fixed threshold. Through iterative processing, the original trajectory is simplified. Various innovative algorithms have been developed based on this approach. Cui et al. constructed a compression method for ship space-time AIS trajectory data based on the DP algorithm, selecting multiple thresholds to compress the upbound and down-bound ship AIS trajectories separately [15]. Zhou et al. proposed a multi-objective peak DP algorithm that adopts a peak sampling strategy, considers three optimization objectives for the trajectory, and adds an obstacle detection mechanism to create a compression algorithm more suitable for curved trajectories [16]. Yang et al. introduced an Adaptive Douglas-Peucker with Acceleration algorithm, enhancing the compression process by considering both acceleration and distance to a baseline [17].

Another simplification algorithm is the Sliding Window (SW) algorithm, which starts from the initial point of the trajectory, initializes a fixed-size window, and gradually inputs trajectory points in a data stream manner, achieving online compression through a step-by-step approach [18]. Various innovative algorithms have been developed in this area. Liu et al. proposed a novel online vessel trajectory compression method based on the Improved Open Window (IOPW) algorithm, which compresses vessel trajectories instantly according to vessel coordinates along with a timestamp driven by the AIS data [19]. SONG et al. proposed a two-stage online compression algorithm (DTGO), which combines a dynamic threshold value with global optimization. In the first stage, the original trajectory is processed in segments, and the threshold values are dynamically updated, resulting in a simplified trajectory. In the second stage, the simplified trajectory is globally optimized using a modified SPM algorithm [20].

Most of the above AIS simplification algorithms have several limitations: they require clear starting and ending points and work effectively with data from a single journey or a segment of a complex route [21]. Additionally, they need to utilize speed, heading, and navigation status information provided by AIS data, alongside position information, to achieve good results for high-quality AIS trajectory data. However, their effectiveness diminishes with AIS data lacking attribute information. Moreover, these simplification algorithms do not segment trajectories into multiple independent routes, which is necessary to reflect the reality where ships navigate according to several different trajectories. Therefore, there is a need to develop more effective trajectory data simplification algorithms that consider the unique characteristics of these trajectories.

## Our method

### Problem formulation

To obtain typical routes using AIS data, the first step is to capture the trajectories formed by this data. AIS data, being cumulative, inherently includes potential temporal characteristics due to its chronological order. During the research process, a subset of the data is typically extracted, and by connecting the data points sequentially, a trajectory can be formed for analysis, as illustrated in Fig 1.

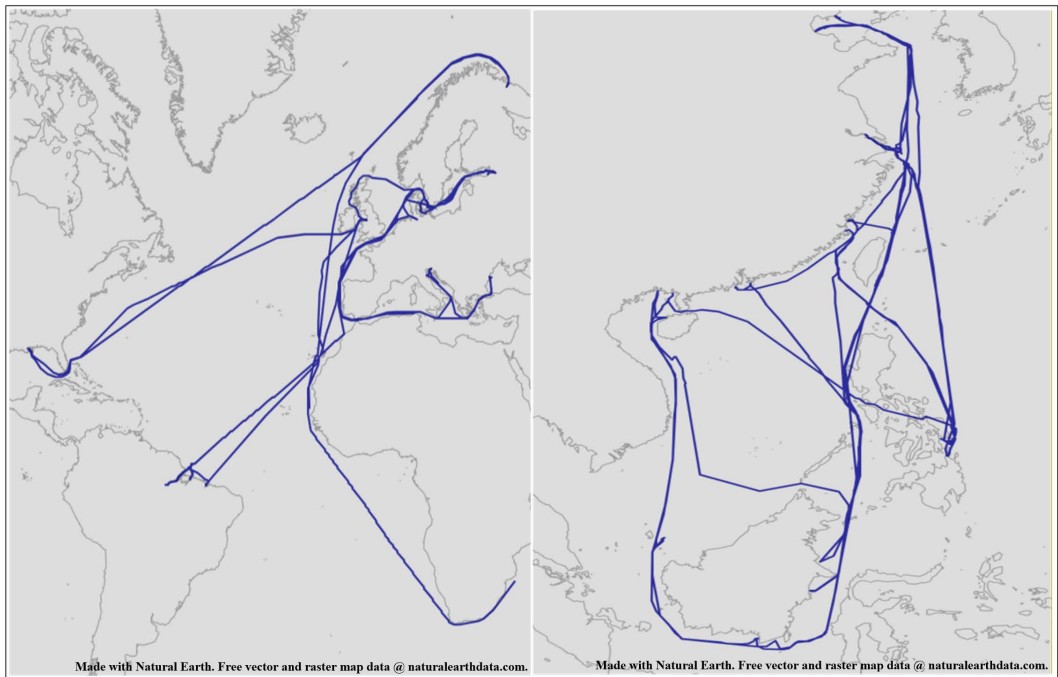

**Fig 1. AIS trajectories of a ship, showing complexity, repetition, and some redundancy.** This figure was generated using public domain map data from Natural Earth (http://www.naturalearthdata.com/).

As observed in Fig 1, these trajectories exhibit several characteristics: First, they are cyclic and overlapping, with no clear starting or ending points. Second, certain areas of the trajectories are disorganized and highly redundant. Third, the overall trajectory may contain multiple routes, as depicted in Fig 2.

## Overview of the method

To effectively extract typical routes from AIS data, we propose an improved typical route extraction algorithm based on AIS data simplification and motion continuity. The algorithm comprises two main parts:

1. Simplification of Complex Trajectories Based on Data Continuity: The raw AIS data is input into the algorithm, which simplifies the trajectory using a distance threshold and segments it into multiple simplified routes. However, at this stage, issues such as deviations in starting points and inaccurate segmentation may arise.

2. Trajectory Optimization Based on Ship Navigation Patterns: The simplified routes are input into the algorithm, which optimizes the starting points of the previously obtained trajectory segments using navigation direction consistency. The algorithm also merges inaccurately segmented routes to form accurate typical routes for the ship's navigation.

The entire process, including both simplification and optimization steps, is illustrated in Fig 3.

## Simplification of complex trajectories based on data continuity

We denote the raw AIS data as $P_{raw\_data} = \left\{ \left[x_0, y_0\right], \left[x_1, y_1\right], ..., \left[x_n, y_n\right] \right\}$. To calculate the distance between two adjacent points, we use the Haversine formula [22]. The Haversine formula is a spherical trigonometric function commonly used to calculate the great-circle distance between two points on the Earth's surface, given their latitude and longitude. The specific formula is as follows:

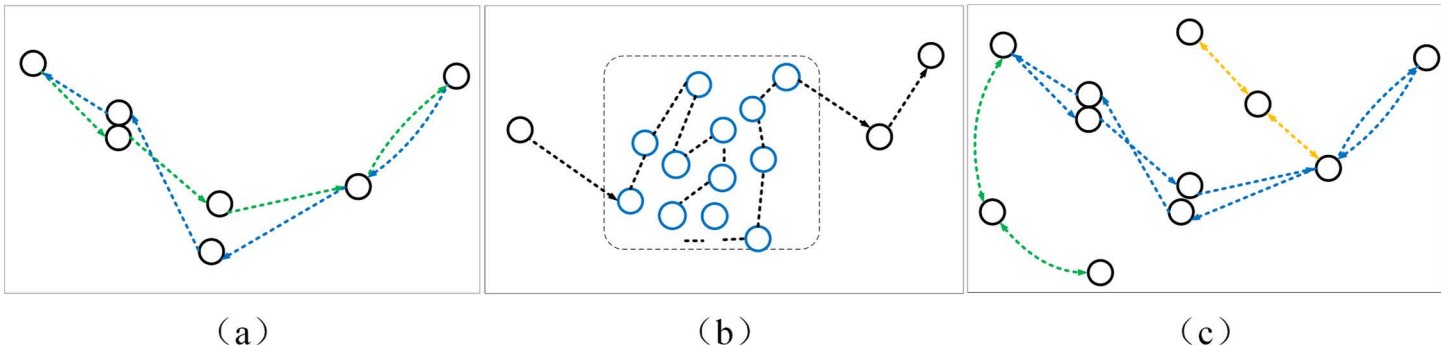

**Fig 2. Current state of trajectories based on AIS data.** (a) Trajectories are cyclic and lack clear start or end points; (b) Trajectories contain complex and repetitive sections; (c) Trajectories may include multiple routes encompassing both (a) and (b) scenarios.

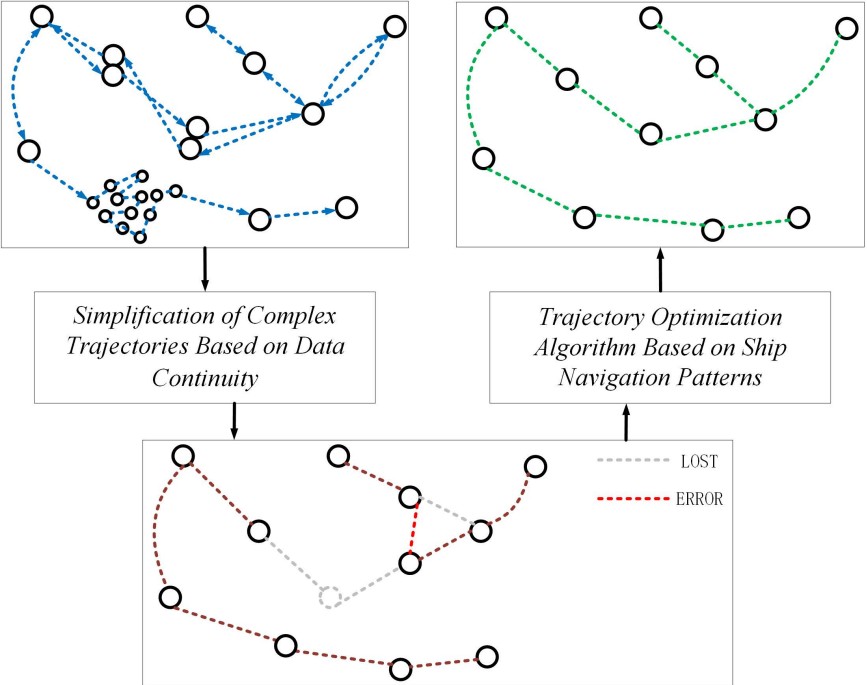

**Fig 3. Workflow of the proposed typical route extraction framework.** The process comprises two sequential stages. Stage 1 applies a distance threshold to raw AIS data, generating simplified candidate routes. These routes feed directly into Stage 2, which refines them by: (a) correcting deviated starting points using route direction continuity, and (b) merging segments based on endpoint directional similarity and proximity.

$$a = \sin^2\left(\frac{x_n - x_{n-1}}{2}\right) + \cos x_n \cos x_{n-1} \sin^2\left(\frac{y_n - y_{n-1}}{2}\right) \tag{1}$$

$$c = 2 * \arctan 2\left(\frac{\sqrt{a}}{\sqrt{1-a}}\right) \tag{2}$$

Using the Haversine formula, we can calculate the angular distance between points $(x_{n-1}, y_{n-1})$ and $(x_n, y_n)$. The arc length distance $D$ between the two points is then computed using and the Earth's radius $R$ ($R \approx 6371\,\text{km}$), with the following formula:

$$D = R * c = 2R * \arctan 2 \left( \frac{\sqrt{a}}{\sqrt{1-a}} \right)$$

(3)

We will use a custom parameter $D_{threshold}$ to filter the necessary AIS coordinate points. Certain assumptions are necessary: we assume that the raw data $P_{raw\_data}$ may contain $s$ routes. We denote these routes collectively as $P_{route}$, where $P_{route} = \{ [R_0], [R_1], ..., [R_s] \}$ and $R_s = [x_0, y_0], [x_1, y_1], ..., [x_m, y_m]$.

First, take the first point from $P_{raw\_data}$ as the starting point and assign it into the first trajectory $P_{route}$, thus $P_{route} = \{ [[x_0, y_0]] \}$. Then, sequentially consider each subsequent point $p_r \in P_{raw\_data}$ and find the point $p_m$ in $P_{route}$ that is the closest to the current point $p_r$, with $p_m$ being on the trajectory $R_z$ in $P_{route}$.

Next, calculate the minimum distance $D_{min}$ between $p_r$ and $p_m$, and use the distance threshold $D_{threshold}$ to determine whether to retain the point. If $D_{min} < D_{threshold}$, then $p_r$ is not retained; if $D_{min} \geq D_{threshold}$, consider the position of $p_m$ in the trajectory $R_z$. If $p_r$ is the last point, place $p_r$ at the end of $R_z$; if $p_r$ is not the last point, consider $p_m$ as a temporary starting point of a new trajectory and place $p_r$ as the first waypoint of the subsequent route.

Finally, this process transforms $P_{raw\_data}$ into $P_{route}$, which comprises several routes. The specific formula is as follows:

$$P_{route} = \begin{cases} \{ [R_0], [R_1], ..., [\mathbf{R_z}] + [[\mathbf{p_r}]], ..., [R_s] \}, D_{min} \geq D_{threshold} \text{ and } m = len(R_z) \\ \{ [R_0], [R_1], ..., [R_s], [[\mathbf{p_m}, \mathbf{p_r}]] \}, D_{min} \geq D_{threshold} \text{ and } m < len(R_z) \\ \{ [R_0], [R_1], ..., [R_s] \}, D_{min} < D_{threshold} \end{cases}$$

(4)

The distance-based judgment in Eq 4 yields distinct route segmentation outcomes depending on the spatial relationship between the current point $p_r$ and existing routes. When $D_{min} \geq D_{threshold}$ and the nearest point $p_m$ is the endpoint of its route ($m = len(R_z)$), a new route segment is created with $p_r$ as its start, capturing a true divergence. If $D_{min} \geq D_{threshold}$ but $p_m$ is not an endpoint ($m < len(R_z)$), the algorithm branches a new segment from $p_m$, preserving alternative paths or shortcuts. Conversely, if $D_{min} < D_{threshold}$, the point $p_r$ is discarded to avoid redundancy, thereby maintaining a compact representation of the current route. This mechanism ensures that only significant deviations initiate new candidate routes.

After processing the raw AIS data using the complex trajectory simplification algorithm based on data continuity, the data can be segmented into several groups of routes according to distance. However, relying solely on the nearest distance to determine a starting point may cause deviations in route starting points. Additionally, due to the threshold setting, originally continuous routes might be divided into multiple segments.

## Simplification of complex trajectories based on data continuity

Currently, we have obtained several route segments $P_{route}$. We extract the starting points: $P_{start} = \{ [x_0^0, y_0^0], [x_0^1, y_0^1], ..., [x_0^n, y_0^n] \}$ and the ending points $P_{end} = \{ [x_n^0, y_n^0], [x_n^1, y_n^1], ..., [x_n^n, y_n^n] \}$ of these route segments. We will use $P_{start}$ and $P_{end}$ combined with ship navigation patterns to optimize the existing routes and address the issues of starting point deviation and erroneous route splitting.

First, we define two distance parameters $P_{start}$ and $P_{end}$ for the starting points and ending points, respectively. The formulas are as follows:

$$R_{start} = D_{threshold} + C$$

(5)

$$R_{start} = 2D_{threshold} + C$$

(6)

During the initial trajectory simplification, a new route segment is created when a point $p_r$ is beyond $D_{threshold}$ from its nearest point $p_m$ on an existing route, and $p_m$ is not the endpoint. This means the vessel's trajectory has diverged by a distance of at least $D_{threshold}$ to form a new branch. For an ending point of one segment to be considered for connection with another segment $R_z$, it must find a valid connection point on $R_z$. However, the point that caused the original divergence likely lies between $D_{threshold}$ and $2D_{threshold}$ from $R_z$. Therefore, setting $R_{start} = 2D_{threshold} + C$ (where $C$ is a small constant to account for spherical distance calculation uncertainties) defines a search radius that is sufficiently large to encompass the potential divergence area and rediscover the original route connection point, yet constrained enough to avoid spurious connections with unrelated routes.

The design of $P_{start}$ and $P_{end}$ is illustrated in Fig 4. The parameter $D_{threshold}$ controls the inclusion of AIS points into a route by defining the connection between the starting point and the subsequent points. The formation of a new route requires that its starting point is at least $D_{threshold}$ away from an existing route. Similarly, the ending point is defined by this threshold, but reconnecting it to an unrelated route requires a search radius of $2 \times D_{threshold}$ to account for the initial divergence. We thus construct circles $C_{start}$ and $C_{end}$ around the points in $P_{start}$ and $P_{end}$ with radii $R_{start}$ and $R_{end}$, respectively.

Next, we assume $p_0^m = \left[ x_0^m, y_0^m \right] \in R_m$ and determine whether the circle $C_{start}^m$ formed by $p_0^m$ intersects with other trajectories in $P_{route}$ or if $p_0^m$ itself is a point on other trajectories. Suppose this trajectory is $R_n$. Let $p_1^m, p_2^m \in R_m$ be the two points after $p_0^m$. As previously mentioned, the connection between $p_0^m$ and $p_1^m$ only used the nearest principle, which can only prove that $R_n$ and $R_m$ may intersect. Considering that sudden changes in turning angles do not occur in standard routes [23], it is necessary to readjust the intersection point $p_0^m$ of $R_n$ and $R_m$. Based on points $p_1^m$ and $p_2^m$, draw a ray (or great circle route) with $p_2^m$ as the vertex and $p_2^m$ pointing towards $p_1^m$, intersecting $R_n$ at $p_{intersect-s}$. Find the point $p_{nearest-s}$ on $R_n$ closest to $p_{intersect-s}$, so $R_m = \left[ p_{nearest-s} \right] + \left[ p_1^m, p_2^m, ..., p_n^m \right]$. This corrects the starting point deviation, as shown in Fig 5.

Then, obtain the ending point $p_{n0}^m = \left[ x_n^m, y_n^m \right] \in R_m$. Similarly, determine whether the circle $C_{end}^m$ formed by $p_n^m$ intersects with other trajectories in $P_{route}$ or if $p_n^m$ itself is a point on other trajectories. Suppose this trajectory is $R_z$. Let $p_{n-1}^m \in R_m$ be a point before $p_n^m$. Based on points $p_n^m$ and $p_{n-1}^m$, draw a ray (or great circle route) with $p_{n-1}^m$ as the vertex and $p_{n-1}^m$ pointing

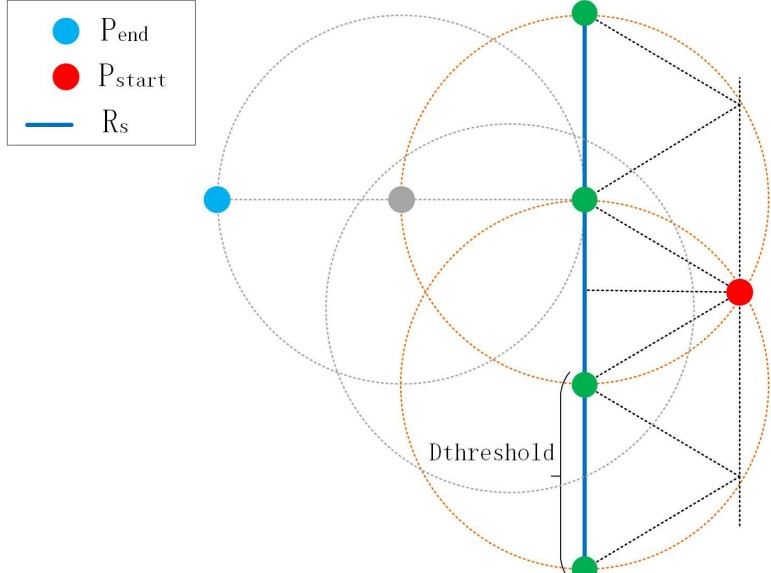

**Fig 4. Diagram Showing the Distance Between the Starting and Ending Points and the Route.** This diagram illustrates the search radii $R_{start}$ and $R_{end}$, which control the connection of route segments. $R_{start}$ corrects deviated starting points by linking them to nearby routes, while the larger $R_{end}$ reconnects incorrectly split segments by spanning the potential divergence region between them.

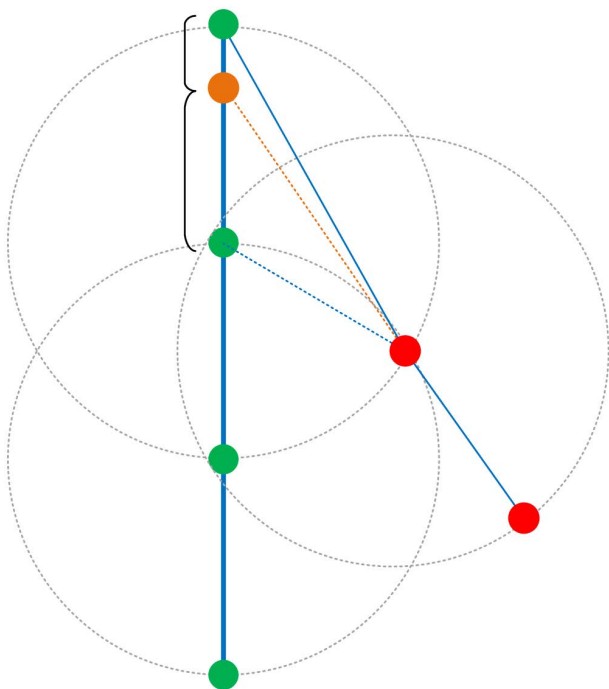

**Fig 5. Diagram Illustrating the Adjustment of Starting Points Using Route Direction.** The algorithm projects the initial direction of the route backwards. This projection intersects an existing route at $p_{nearest-s}$. The point $p_{nearest-s}$ on that route is then identified and becomes the new, corrected starting point, ensuring topological continuity.

towards $p_n^m$, intersecting $R_z$ to obtain the intersection point $p_{intersect-e}$. Find the point $p_{nearest-e}$ on $R_z$ that is closest to $p_{intersect-e}$, so $R_m = [p_0^m, p_1^m, ..., p_n^m] + [p_{nearest-e}]$. Then, merge the route segments with the same starting and ending points.

This way, a new set of routes $P_{new-route}$ with new starting points $P_{new-start}$ and ending points $P_{new-end}$ can be obtained. Again, construct circles $C_{new-start}$ with points in $P_{new-start}$ as the centers and radius $R_{start}$, and circles $C_{new-end}$ with points in $P_{new-end}$ as the centers and radius $R_{end}$. Determine whether $C_{new-start}$ intersects with $C_{new-end}$. If they intersect, we consider the routes with the starting and ending points to be potentially related. Taking into account that sudden changes in turning angles do not occur in standard routes, further judgment can be made based on the directional similarity and consistency of the two routes to determine whether they can be connected, as shown in Fig 6.

As observed in Fig 6, the angle $\theta_{route}$ between the vector of the route direction $\overrightarrow{BA}$ at the starting point and the vector of the route direction $\overrightarrow{DC}$ at the ending point is calculated using the following formula:

$$\theta_{route} = \arccos\left(\frac{(x_1^z - x_0^z)(x_n^m - x_{n-1}^m) + (y_1^z - y_0^z)(y_n^m - y_{n-1}^m)}{\sqrt{(x_1^z - x_0^z)^2 + (y_1^z - y_0^z)^2} \cdot \sqrt{(x_n^m - x_{n-1}^m)^2 + (y_n^m - y_{n-1}^m)^2}}\right)$$

(7)

Finally, calculate the angle $\theta$ between the tangent of the great circle route formed by points A and B at point B and the straight line $\overrightarrow{BA}$ using the following formula:

$$\theta = \arccos\left(\sin(x_0^z)\sin(x_1^z) + \cos(x_0^z)\cos(x_1^z)\cos(y_1^z - y_0^z)\right)$$

(8)

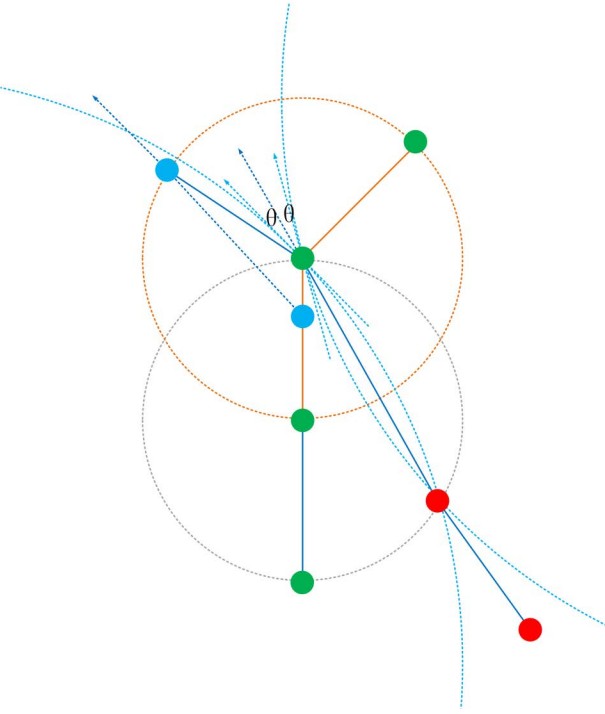

**Fig 6. Diagram illustrating the determination of route association based on the similarity of starting and ending point directions.** Two segments are merged if the angle $\theta_{route}$ between their respective direction vectors ($\overrightarrow{BA}$ and $\overrightarrow{DC}$) falls below a threshold $\theta$, ensuring directional continuity in the resulting typical route.

If $\theta_{route} < \theta$, it can be considered that the directions of the two route segments are similar and consistent, and the distances are close. Therefore, the two route segments can be merged into one by removing the ending point of the route and connecting it to the starting point of the next route. The final new route is denoted as $R_m = \left[ p_0^m, p_1^m, ..., p_{n-1}^m \right] + R_z$.

By processing multiple route segments using the trajectory optimization algorithm based on ship navigation patterns, the issues of route starting point deviation and route segment continuity can be effectively addressed.

## Experiments

### Datasets

Currently, there is a limited amount of open-source AIS data available, and bulk access to AIS data typically requires application and purchase from specialized agencies. Most open-source AIS datasets often contain limited information and may only cover a single voyage or a segment of a route, which fails to accurately reflect actual ship navigation conditions.

For our experiments, we will use AIS data collected from 10 ships over the past 5 years, comprising a total of 420,000 records. This data is recorded every 8 minutes and includes only the ships' latitude and longitude coordinates, along with the recording time. It has not been filtered or cleaned, allowing it to accurately reflect the characteristics of raw AIS data, as illustrated in Fig 7.

### Results of experiments

Using the complex trajectory simplification algorithm based on data continuity, the raw AIS data is processed to obtain multiple route segments, as shown in Fig 8.

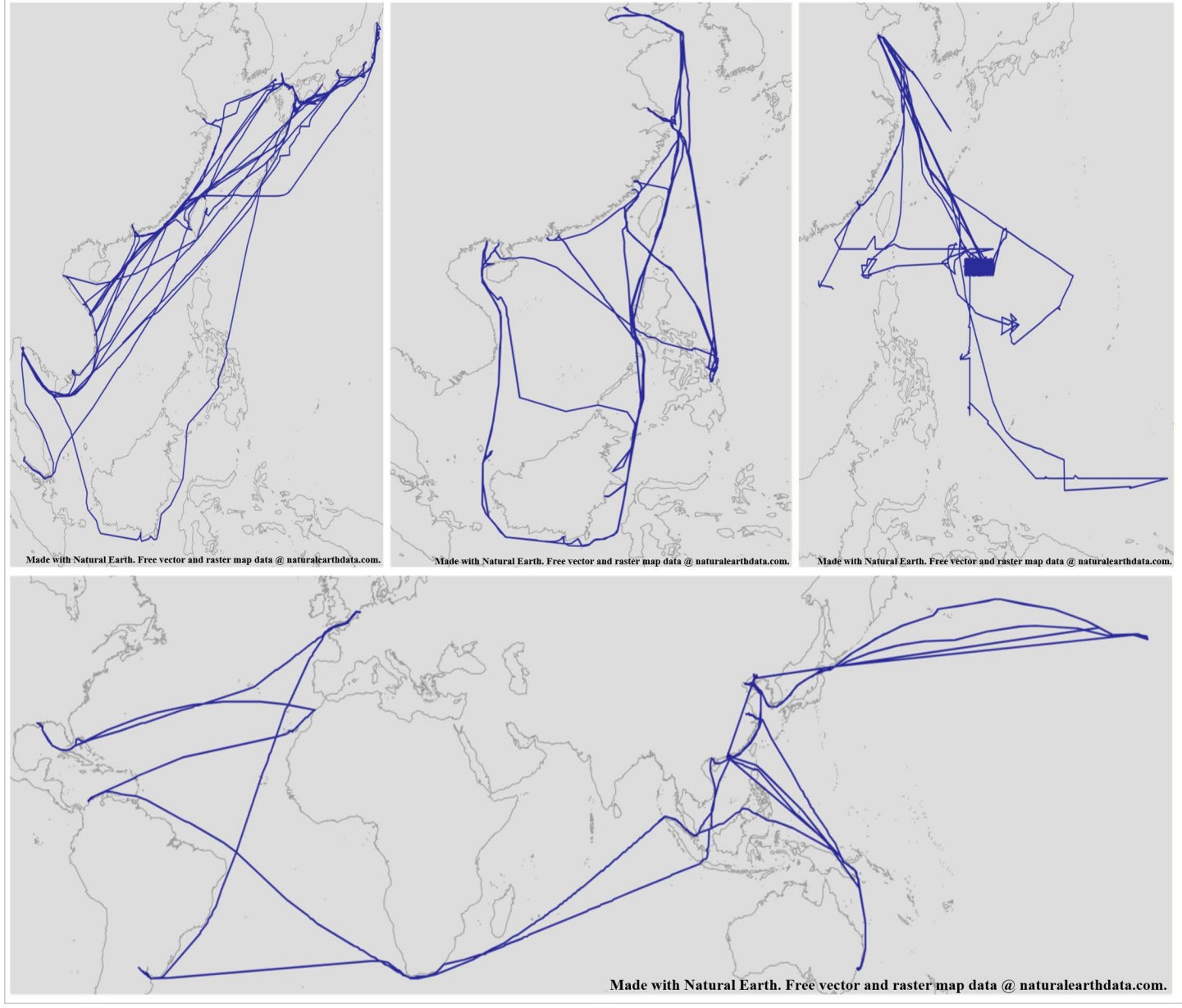

**Fig 7. Trajectory Diagram Formed by AIS Data of Ships Over the Past Five Years.** This figure was generated using public domain map data from Natural Earth (http://www.naturalearthdata.com/).

From Fig 8, it can be seen that after processing the raw AIS data using the complex trajectory simplification algorithm based on data continuity, the data volume is reduced, and the originally complex trajectory lines become simplified and intuitive. Additionally, the trajectory lines can be divided into multiple possible routes. The details of the processed routes are shown in Fig 9.

From Fig 9, it can be seen that there are instances where starting and ending points are not related to any other route segments, and cases where points that should be related are separated. However, using the circles constructed by the

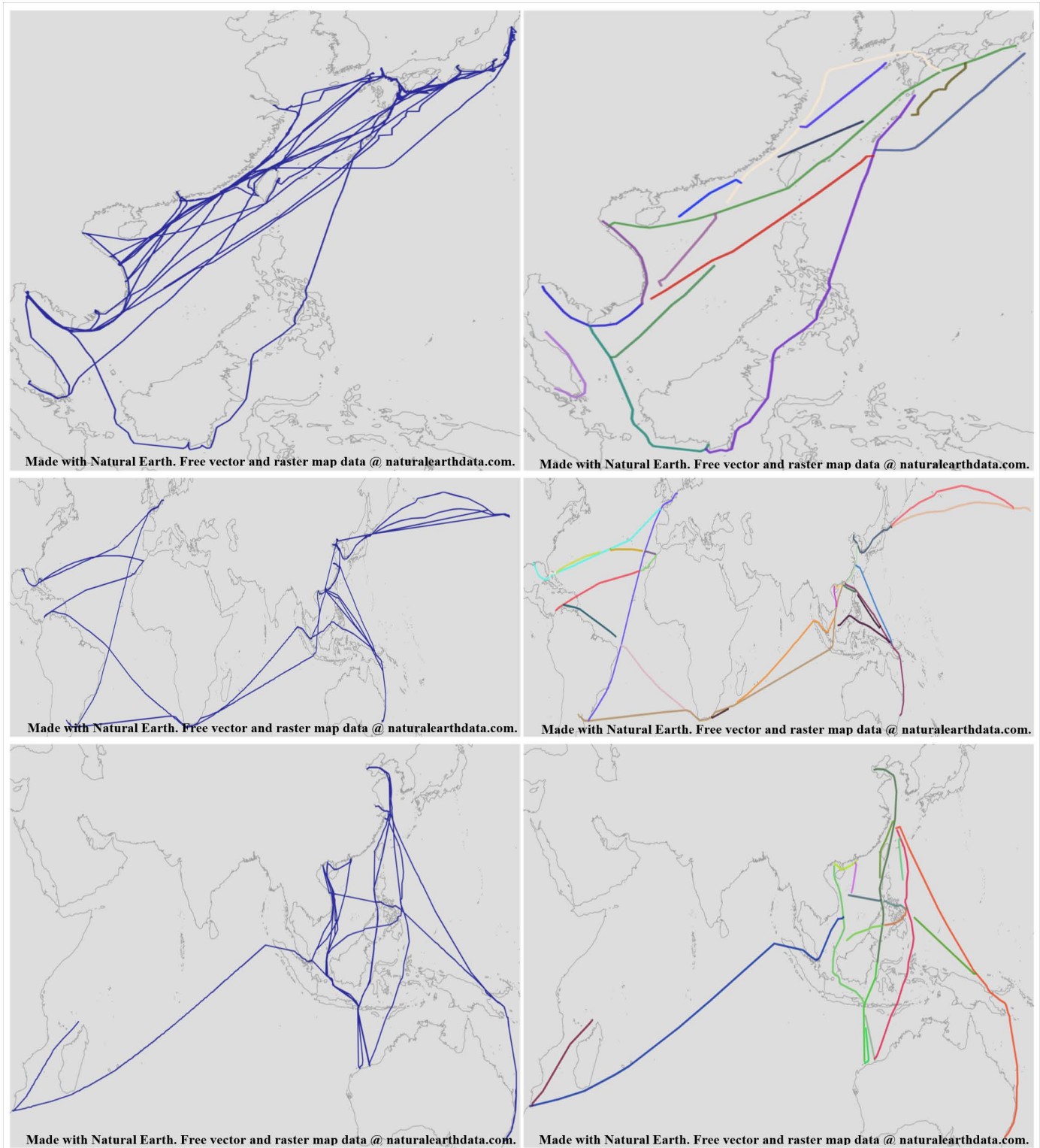

**Fig 8. Comparison Between Raw Data and Data Processed by the Complex Trajectory Simplification Algorithm Based on Data Continuity.** This figure was generated using public domain map data from Natural Earth (http://www.naturalearthdata.com/).

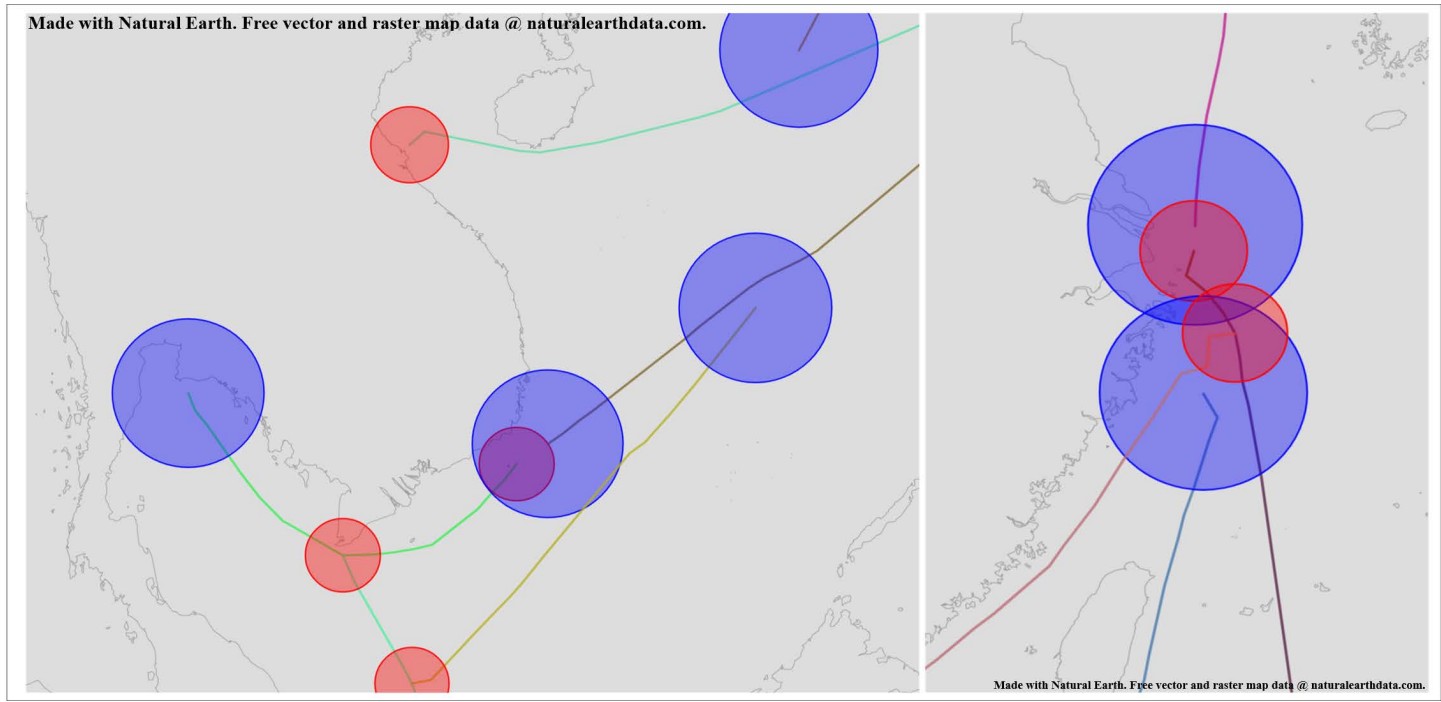

**Fig 9. Detailed Diagram of Data Processed by the Complex Trajectory Simplification Algorithm Based on Data Continuity.** This figure was generated using public domain map data from Natural Earth (http://www.naturalearthdata.com/).

starting and ending points as designed in the previous algorithm, we can determine the actual associated routes. Therefore, it is necessary to use the trajectory optimization algorithm based on ship navigation patterns designed in this paper to process the route segments and obtain the optimized routes, as shown in Fig 10.

From Fig 10, it can be seen that after processing the route segments using the trajectory optimization algorithm based on ship navigation patterns, the details of the processed routes are shown in Fig 11.

From Fig 11, it can be seen that through the intersection and extension direction judgment of the starting and ending points with other routes, the originally deviated starting points are adjusted to be accurate, and the originally discontinuous ending points are extended forward to find possible intersection points. By using the direction similarity and consistency of the starting and ending points, the route connections become coherent. The algorithm proposed effectively simplifies AIS data to form shipping routes, with the simplification results shown in Table 3.

As seen from Table 1, our method outperforms the DP algorithm and the SW algorithm in terms of compression rate. This is because the primary goal of our algorithm is to extract the typical features of ship navigation routes, which involves pruning the original route and may not fully preserve its basic shape characteristics. In contrast, the DP and SW algorithms are capable of maintaining the basic shape characteristics of the original route but are unable to extract typical route features. Consequently, they also retain redundant and repetitive aspects of the original route.

## Trajectory prediction and anomaly detection experiments

In addition to the typical route extraction described earlier, we conducted further experiments to evaluate the effectiveness of our simplification algorithm in the context of deep learning applications, specifically trajectory prediction and anomaly detection. These experiments aim to demonstrate the advantages of the proposed algorithm in improving both the efficiency and accuracy of downstream tasks.

   

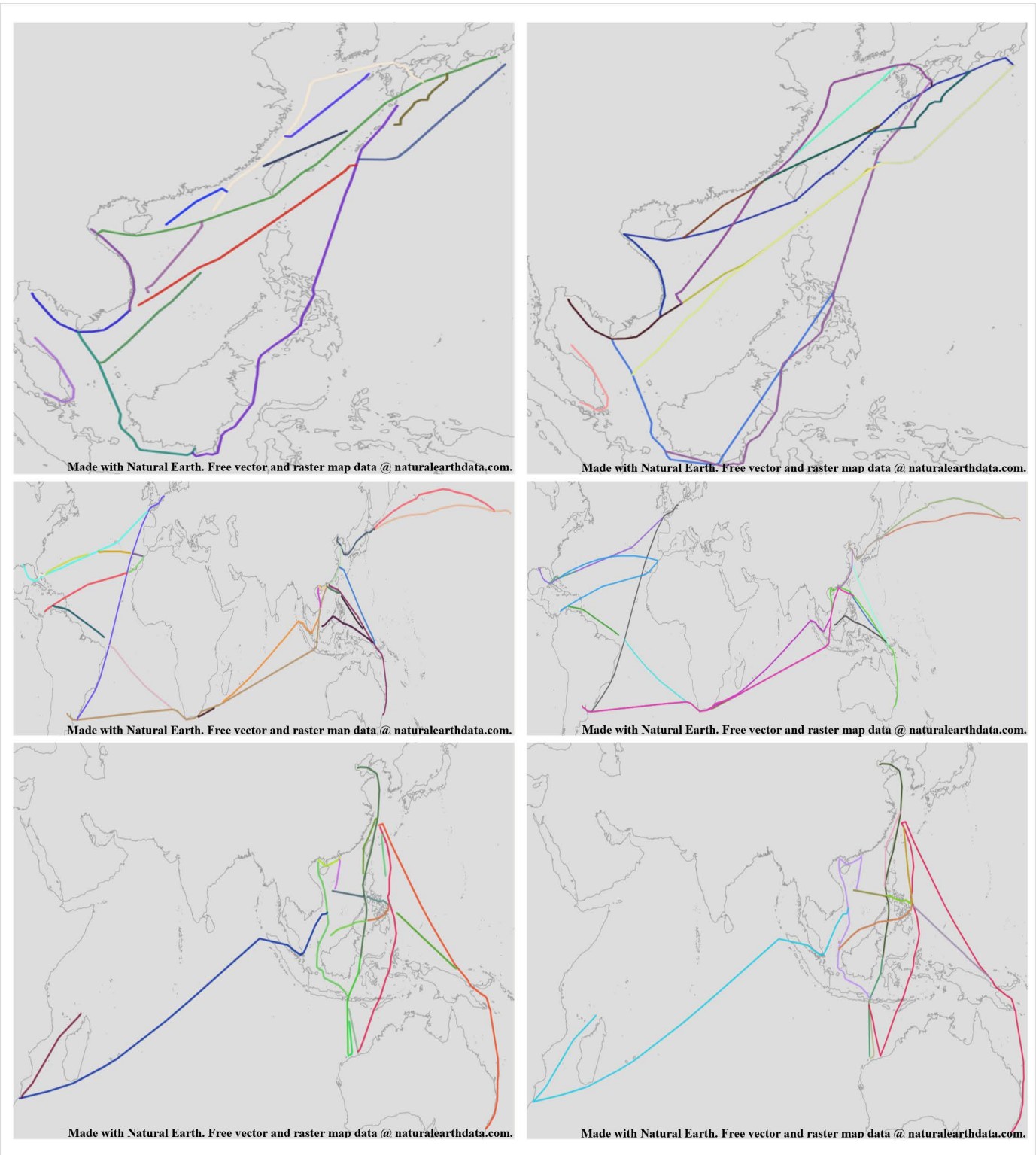

**Fig 10. Comparison Between Thinned Routes and Data Processed by the Trajectory Optimization Algorithm Based on Ship Navigation Patterns.** This figure was generated using public domain map data from Natural Earth (http://www.naturalearthdata.com/).

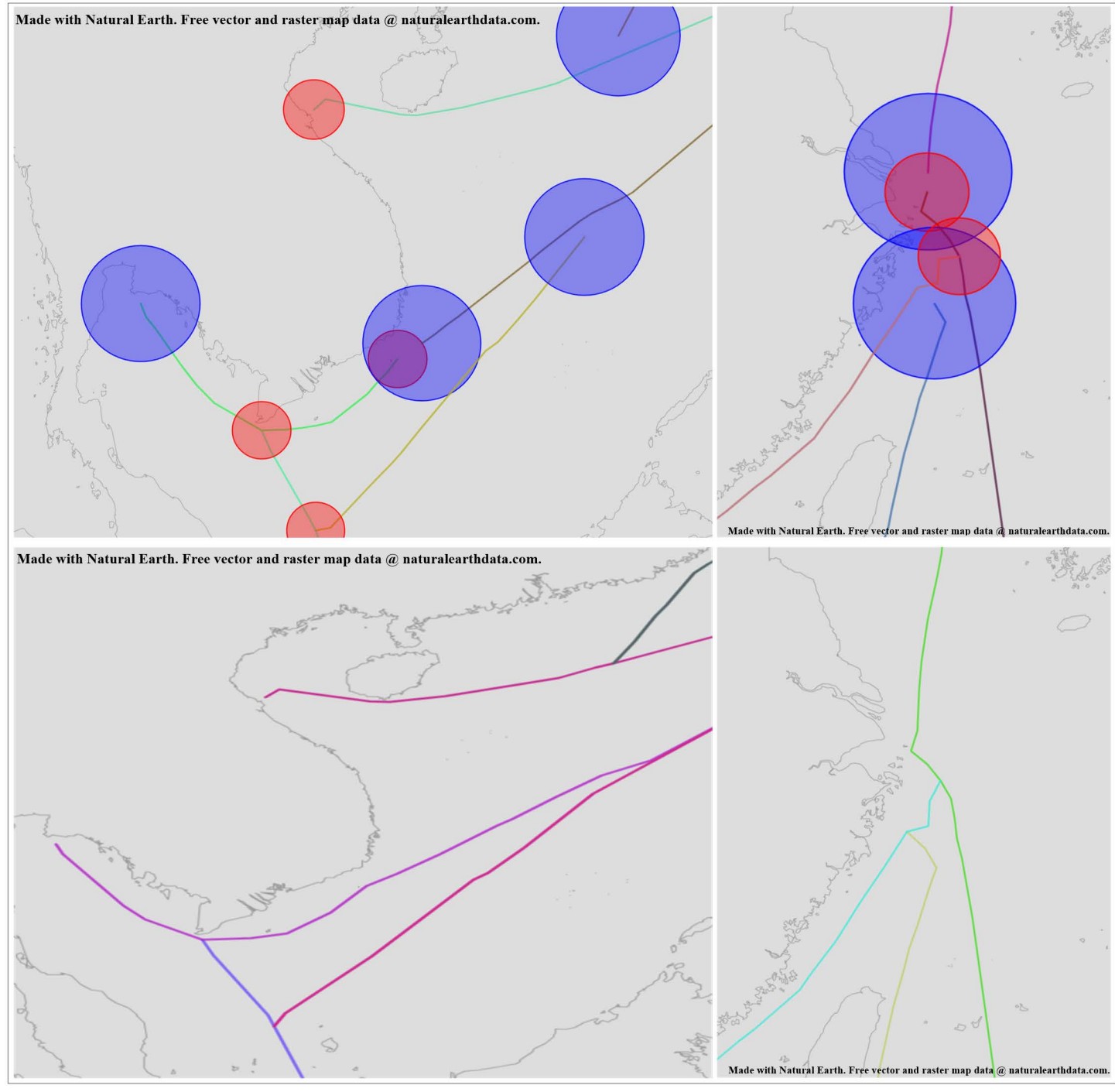

**Fig 11. Detailed Diagram of Trajectories Processed by the Trajectory Optimization Algorithm Based on Ship Navigation Patterns.** This figure was generated using public domain map data from Natural Earth (http://www.naturalearthdata.com/).

**Table 1. The compression ratio use DP algorithm, SW algorithm and our algorithm.**

| Route ID | Raw | DP Algorithm | | SW Algorithm | | Our Algorithm | |
|---|---|---|---|---|---|---|---|
| | | Compressed | Compression Ratio | Compressed | Compression Ratio | Compressed | Compression Ratio |
| 1 | 46354 | 466 | 99.0% | 1355 | 97.1% | 371 | 99.2% |
| 2 | 55203 | 792 | 98.6% | 1682 | 97.0% | 593 | 98.9% |
| 3 | 55414 | 864 | 98.4% | 1196 | 97.8% | 312 | 99.4% |
| 4 | 18754 | 496 | 97.4% | 698 | 96.3% | 108 | 99.4% |
| 5 | 26225 | 805 | 96.9% | 1025 | 96.1% | 93 | 99.6% |
| 6 | 45596 | 784 | 98.3% | 892 | 98.0% | 343 | 99.2% |
| 7 | 6039 | 182 | 97.0% | 192 | 96.8% | 78 | 98.7% |
| 8 | 50875 | 350 | 99.3% | 1468 | 97.1% | 263 | 99.5% |
| 9 | 80739 | 799 | 99.0% | 2855 | 96.5% | 1204 | 98.5% |
| 10 | 17145 | 115 | 99.3% | 54 | 99.7% | 33 | 99.8% |
| Mean | / | / | 98.32% | / | 97.24% | / | 99.23% |

We applied three widely-used deep learning models for trajectory prediction, LSTM,GRU and Transformer, on both the raw AIS data and the simplified typical route data. The objective was to predict future vessel positions based on historical trajectory data.For a consistent comparison, all models were implemented under a common framework. The LSTM and GRU models each comprised two recurrent layers with a hidden size of 256 units. The Transformer model was configured with a comparable capacity, featuring a 128-dimensional embedding size and 4 attention heads across its encoder-decoder architecture. All models were trained for 200 epochs using the Adam optimizer with a learning rate of 0.001 and a batch size of 64.

We evaluated the models using Mean Squared Error (MSE), Root Mean Squared Error (RMSE), and training time. The results, as summarized in Table 2, show that the simplification algorithm significantly improves both training efficiency and predictive accuracy. For LSTM, GRU and Transformer models, the simplified data led to lower RMSE values and shorter training times compared to the original data, demonstrating that the algorithm effectively reduces noise and redundancy, enabling the models to learn faster and more accurately.

From Table 2, we can see that the training time is significantly reduced when using simplified data, and the prediction accuracy (in terms of MSE and RMSE) is also improved.

We further tested the impact of our data simplification algorithm on anomaly detection tasks, using two different methods: Mahalanobis Distance and Isolation Forest. These methods were applied to detect deviations from typical routes, using both the original and simplified datasets.

The results, presented in Table 3, demonstrate that anomaly detection is more accurate and reliable when using the simplified data. Both Mahalanobis Distance and Isolation Forest achieved higher accuracy, precision, and recall when

**Table 2. Comparison of model performance on raw and simplified AIS data.**

| Dataset | Model | MSE | RMSE | Training Time (s) |
|---|---|---|---|---|
| Original Data | LSTM | 0.010 | 0.100 | 320 |
| Simplified Data | LSTM | 0.007 | 0.084 | 240 |
| Original Data | GRU | 0.012 | 0.110 | 300 |
| Simplified Data | GRU | 0.008 | 0.089 | 220 |
| Original Data | Transformer | 0.005 | 0.068 | 340 |
| Simplified Data | Transformer | 0.004 | 0.054 | 260 |

**Table 3. Performance comparison of anomaly detection methods on original and simplified datasets.**

| Dataset | Method | Accuracy | Precision | Recall |
|---|---|---|---|---|
| Original Data | Mahalanobis Dist. | 89% | 85% | 80% |
| Simplified Data | Mahalanobis Dist. | 92% | 88% | 85% |
| Original Data | Isolation Forest | 91% | 87% | 84% |
| Simplified Data | Isolation Forest | 94% | 91% | 89% |

applied to the simplified dataset. This is largely due to the reduction of noise and redundant trajectory data, which allowed for more precise identification of true anomalies.

The simplified data not only improved detection accuracy but also reduced false positives, especially when using the Isolation Forest method. The results show that the proposed algorithm is effective in filtering out irrelevant data, allowing for more accurate and efficient anomaly detection.

## Discussion

We proposed a complex trajectory simplification algorithm based on data continuity and a trajectory optimization algorithm based on ship navigation patterns. These algorithms were validated through multiple sets of experiments, with the results displayed both macroscopically and in detail through comparative graphs. The trajectory simplification algorithm can divide the original AIS data into multiple candidate routes, but there are some issues with the details of these candidate routes. By applying the trajectory optimization algorithm, these candidate routes can be transformed into representative typical routes, effectively improving existing problems.

Furthermore, our use of a distance threshold to filter AIS data results in data compression, as evidenced by the compression rate comparison table, which shows that our algorithm achieves a high compression rate. Unlike other AIS data compression algorithms, the goal of our algorithm is to extract the main features of the routes and eliminate redundant trajectories. Given that actual AIS data contains many redundant trajectories, our algorithm achieves a higher compression rate.

The experiments on trajectory prediction and anomaly detection demonstrate the significant advantages of the proposed simplification algorithm. The reduction of redundant and noisy data points leads to faster model training and higher predictive accuracy, without sacrificing key information about vessel movement patterns. Moreover, the improvement in anomaly detection metrics underscores the algorithm's ability to enhance maritime safety applications by providing a clearer representation of typical vessel routes.the simplification algorithm not only improves the computational efficiency of deep learning models but also enhances their accuracy in both predictive and anomaly detection tasks.

Through multiple rounds of experiments, we demonstrate that our algorithm is capable of processing trajectories formed from real AIS data, transforming these trajectories into typical routes for subsequent research use. However, several limitations of our approach should be acknowledged. First, the performance of our algorithm is sensitive to the distance threshold parameter, whose optimal setting may require empirical tuning for different application scenarios. Second, our current method primarily relies on positional data and motion continuity, while other valuable AIS attributes such as vessel type and navigation status have not been fully utilized. These limitations point to promising directions for future work, including developing adaptive threshold selection methods and incorporating multi-dimensional AIS information to achieve more intelligent route extraction.

## Conclusion

We have proposed a algorithm for converting AIS data into typical route data, leveraging distance threshold filtering and ship navigation patterns. This approach is specifically tailored for extracting representative routes from cumulative data, a

task that differs from the objective of traditional trajectory compression. Our method begins with complex AIS data, applying a distance threshold to simplify the data and process it into multiple potential route segments. We then optimize these segments using ship navigation routes. Unlike other simplification algorithms, our method can address the real-world situation where AIS data may include multiple routes, allowing us to segment the data and merge possible route segments based on directional similarity and consistency.

Additionally, AIS data can be categorized by season or other characteristics, and our method can analyze route changes across different seasons, thereby identifying actual navigation patterns and effectively supporting route planning. Our algorithm can also be integrated with weather data to ascertain the meteorological factors influencing route changes, providing a foundation for meteorological navigation in ship routing.

## Acknowledgments

We sincerely thank our research team colleagues for their valuable discussions and technical support during data processing and algorithm optimization, as well as Dalian Maritime University and the China Waterborne Transport Research Institute for their joint guidance and computational resources support.

## Author contributions

**Data curation:** Jinjia Ruan.

**Formal analysis:** Jin He, Yao Tong.

**Methodology:** Jin He.

**Validation:** Yao Tong.

**Writing – original draft:** Jin He.

**Writing – review & editing:** Jinjia Ruan.

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
