## [Decision Letter · Decision Letter 0]

12 Oct 2025

Dear Dr. Ruan,

Thank you for submitting your manuscript to PLOS ONE. After careful consideration, we feel that it has merit but does not fully meet PLOS ONE’s publication criteria as it currently stands. Therefore, we invite you to submit a revised version of the manuscript that addresses the points raised during the review process.

We look forward to receiving your revised manuscript.

Kind regards,

Dr. Guojin Qin

Academic Editor

PLOS ONE

3. Thank you for uploading your study's underlying data set. Unfortunately, the repository you have noted in your Data Availability statement does not qualify as an acceptable data repository according to PLOS's standards.

6. We note that Figures 1,7,8,9,10 and 11 in your submission contain [map/satellite] images which may be copyrighted. All PLOS content is published under the Creative Commons Attribution License (CC BY 4.0), which means that the manuscript, images, and Supporting Information files will be freely available online, and any third party is permitted to access, download, copy, distribute, and use these materials in any way, even commercially, with proper attribution. For these reasons, we cannot publish previously copyrighted maps or satellite images created using proprietary data, such as Google software (Google Maps, Street View, and Earth). For more information, see our copyright guidelines: http://journals.plos.org/plosone/s/licenses-and-copyright.

1. You may seek permission from the original copyright holder of Figures 1,7,8,9,10 and 11 to publish the content specifically under the CC BY 4.0 license.

Additional Editor Comments (if provided):

Reviewers' comments:

Reviewer's Responses to Questions

**Comments to the Author**

1. Is the manuscript technically sound, and do the data support the conclusions?

Reviewer #1: Yes

Reviewer #2: Yes

2. Has the statistical analysis been performed appropriately and rigorously?

Reviewer #1: Yes

Reviewer #2: Yes

3. Have the authors made all data underlying the findings in their manuscript fully available?

Reviewer #1: Yes

Reviewer #2: Yes

4. Is the manuscript presented in an intelligible fashion and written in standard English?

Reviewer #1: Yes

Reviewer #2: Yes

Reviewer #1: 1. Please clarify the key differences between the proposed algorithm and representative methods such as Sliding Window, IOPW, and DTGO. In both the Introduction and Conclusion, highlight the specific practical problems your method can address that previous approaches could not.

2. The mathematical derivations are not sufficiently systematic and contain logical gaps. For example, the choice of using twice the threshold (Rend = 2Dthreshold + C) lacks theoretical justification.

3. The dataset size and source are insufficient to demonstrate the generalizability of the algorithm. Using only 10 vessels’ AIS data is not representative.

4. Provide the key hyperparameters of the deep learning models (e.g., number of layers, hidden units, learning rate, batch size, training epochs).

5. The explanations of figures (e.g., Fig. 4–6) are too brief, making it difficult for readers to fully understand their meaning.

6. The Discussion section mostly restates experimental results and lacks deeper reflection on the method’s limitations and potential directions for future work.

Reviewer #2: This paper proposes an improved AIS data simplification algorithm designed to extract typical routes from raw AIS data while considering the continuity of vessel motion. The experimental design is reasonable, and the experimental results show that the algorithm has a good compression rate and can improve the trajectory prediction.

Minor revisions are recommended prior to acceptance.

Details Requiring Revision (Suggested Modifications)

1.The core filtering parameter for trajectory simplification lacks explanation of how its value affects compression rate and route segmentation accuracy. Supplement experiments with different values to clarify the optimal parameter range.

2.LSTM, GRU, and Transformer are used in trajectory prediction, but their hyperparameters are unspecified.

3.The two-step algorithm flow figure (e.g., Fig 3) lacks explanation of how "trajectory simplification"-output candidate routes link to the "trajectory optimization" module. Supplement captions or necessary notes.

4.For route segmentation, the paper mentions distance-based judgment of route retention/addition but does not specify specific outcomes under different distance conditions. Add corresponding explanations.

5. Terms such as "starting point" and "initial waypoint" are used interchangeably. It is recommended to standardize the terminology, for instance, consistently using "starting point".

6.Some references lack journal names or volume/issue numbers (e.g., Reference [18]). It is recommended to unify the reference format according to the journal's guidelines.

7. Other Minor Issues:

Page 2: "We proposes" should be corrected to "We propose".

Page 5: "p-m" should likely be "p_m".

A thorough check for spelling and grammar throughout the manuscript is recommended.

The algorithm is practical, with reliable experimental conclusions. Recommended for acceptance after revision.

**Do you want your identity to be public for this peer review?** For information about this choice, including consent withdrawal, please see our Privacy Policy

Reviewer #1: No

Reviewer #2: No

---

## [Author Response · Author response to Decision Letter 1]

30 Nov 2025

We thank the editors and reviewers for their constructive comments. We have carefully addressed all points raised, as detailed below:

We have provided point-by-point responses to all reviewers' comments and uploaded the 'Response to Reviewers' file.

Formatting: The manuscript has been reformatted according to the PLOS ONE style templates.

Code and Data Availability:The minimal dataset and all author-generated code have been deposited in the Zenodo repository and are publicly available under the DOI: https://doi.org/10.5281/zenodo.17568672.

ORCID: The corresponding author’s ORCID iD has been registered and validated in the submission system.

Figure Copyright: All map figures (Figs 1, 7, 8, 9, 10, and 11) have been replaced with maps sourced from Natural Earth（https://www.naturalearthdata.com/downloads/10m-physical-vectors/10m-coastline/）, which are in the public domain, and have been credited with the note: "Made with Natural Earth. Free vector and raster map data @ naturalearthdata.com." .The manuscript downloaded the 10m Coastline Shapefile from Natural Earth(https://www.naturalearthdata.com/downloads/10m-physical-vectors/10m-coastline/), converted it to a GeoJSON format file, and then used Python's folium.GeoJson() to load the file as the base map. Subsequently, AIS data were overlaid onto this map to form the final figures(Figs 1, 7, 8, 9, 10, and 11). All figures are openly available and comply with the Creative Commons Attribution License (CC BY 4.0).

---

## [Decision Letter · Decision Letter 1]

16 Dec 2025

Improved AIS Data Simplification Algorithm for Extracting Typical Routes Considering Motion Continuity

PONE-D-25-36742R1

Dear Dr. Ruan,

We’re pleased to inform you that your manuscript has been judged scientifically suitable for publication and will be formally accepted for publication once it meets all outstanding technical requirements.

Kind regards,

Guojin Qin

Academic Editor

PLOS One

Additional Editor Comments (optional):

Reviewers' comments:

Reviewer's Responses to Questions

**Comments to the Author**

Reviewer #1: (No Response)

Reviewer #2: All comments have been addressed

2. Is the manuscript technically sound, and do the data support the conclusions?

Reviewer #1: (No Response)

Reviewer #2: Yes

3. Has the statistical analysis been performed appropriately and rigorously?

Reviewer #1: (No Response)

Reviewer #2: Yes

4. Have the authors made all data underlying the findings in their manuscript fully available?

Reviewer #1: (No Response)

Reviewer #2: Yes

5. Is the manuscript presented in an intelligible fashion and written in standard English?

Reviewer #1: (No Response)

Reviewer #2: Yes

Reviewer #1: (No Response)

Reviewer #2: In response to the reviewers' comments, the authors have undertaken a comprehensive revision of the manuscript. All critical issues have been resolved, thereby improving the overall quality of the paper. The research is presented clearly, with sound argumentation, bringing the work into alignment with the journal's academic and publication criteria.

**Do you want your identity to be public for this peer review?** For information about this choice, including consent withdrawal, please see our Privacy Policy

Reviewer #1: No

Reviewer #2: No

---

## [Editor Report · Acceptance letter]

PONE-D-25-36742R1

PLOS One

Dear Dr. Ruan,

I'm pleased to inform you that your manuscript has been deemed suitable for publication in PLOS One. Congratulations! Your manuscript is now being handed over to our production team.

Kind regards,

on behalf of

Dr. Guojin Qin

Academic Editor

PLOS One